# Effects of Corn Intercropping with Soybean/Peanut/Millet on the Biomass and Yield of Corn under Fertilizer Reduction

Likun Li [1], Yan Zou [1], Yanhui Wang [1], Fajun Chen [1,*] and Guangnan Xing [2,3,4,*]

1 Department of Entomology, College of Plant Protection, Nanjing Agricultural University, Nanjing 210095, China; 2018202033@njau.edu.cn (L.L.); 2020202034@stu.njau.edu.cn (Y.Z.); 2018102061@njau.edu.cn (Y.W.)
2 National Center for Soybean Improvement, Nanjing Agricultural University, Nanjing 210095, China
3 State Key Laboratory for Crop Genetics and Germplasm Enhancement, College of Agriculture, Nanjing Agricultural University, Nanjing 210095, China
4 Jiangsu Collaborative Innovation Center for Modern Crop Production, Nanjing Agricultural University, Nanjing 210095, China
* Correspondence: fajunchen@njau.edu.cn (F.C.); xinggn@njau.edu.cn (G.X.)

**Abstract:** Corn (*Zea mays* L.) is one of the key grain crops in China. In fields, the two crops of soybean (*Glycine max* L.) and peanut (*Arachis hypogaea* L.), which have nitrogen-fixing capacity (NFC), are generally used to intercrop with corn to improve plant physiology and production ability of corn even under fertilizer reduction. To explore a more scientific and reasonable way to plant corn, and simultaneously reduce the use of chemical fertilizers and pesticides, the impacts of corn intercropping with two NFC crops (including soybean and peanut) and the a non-NFC crop (i.e., millet (*Setaria italica*)) through five planting patterns, including three intercropping patterns (2 corn rows to 2, 3, and 4 NFC-crop rows or 2, 4, and 6 millet rows) and two sole crop patterns of corn and soybean, peanut, or millet under normal (600 kg/ha) and reduced (375 kg/ha) levels of NPK (N:$P_2O_5$:$K_2O$ = 15:15:15) fertilization levels on the activity of N-metabolism-related enzymes in corn rhizosphere soil and corn leaves, and plant biomass and yield of corn were researched in this study. The results showed that fertilizer reduction significantly decreased the plant biomass and grain yield of the sole crop corn. The intercropping type and planting pattern both had significant effects on the activities of N-metabolism-related enzyme of soil alkaline protease (S-ALPT), and glutamine oxoglutarate aminotransferase (GOGAT), glutamate synthetase (GS), and nitrate reductase (NR) in the leaves of corn plants. The intercropping type of corn with soybean through the planting pattern of 2 corn rows to 4 soybean rows significantly improved the activities of N-metabolism-related enzymes in soil and corn leaves even under the fertilizer reduction. The intercropping pattern of corn-soybean was the most beneficial to increase the total nitrogen content in soil and corn leaves. In addition, the intercropping significantly increased the soil microbial diversity under normal fertilizer. Furthermore, fertilizer reduction significantly increased soil microbial diversity of the corn sole crop. Therefore, it is concluded that for corn in intercropping systems, the best and the worst companion crop were, respectively, soybean and millet.

**Keywords:** corn; intercropping type; planting pattern; fertilizer reduction; N-metabolism-related enzymes; plant biomass and grain yield

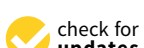

## 1. Introduction

Intercropping means that two or more crops are planted in the same farmland system and closely mixed during the whole or part of a growing season [1], and it has a history of thousands of years in China [2]. As the most representative planting system, intercropping has been widely used in agriculture production because it can enhance the complementarity and utilization of temporal and spatial patterns of nutrients and environmental resources [3,4]. Intercropping can make full use of soil nutrients, enhance crop yield, and

improve soil fertility [5]. For example, the biomass of corn could be increased under the intercropping of corn-velvet bean, which would be increase the yield of corn green feed [6]. It is also possible to alleviate the pressure on land and water resources [7,8]. In conclusion, intercropping increases habitat diversity and has a positive effect on increasing crop yields, increasing species diversity, and reducing pests and diseases [9]. Therefore, intercropping is called the "New Green Revolution" of agricultural sustainability intensification [10]. Moreover, intercropping can also inhibit the occurrence of diseases and insect pests in the agroecosystem [11–13].

Some enzymes present in soil can decompose plant tissues and convert them into nutrients that can be absorbed by roots to improve plant growth [14]. Some studies have shown that the activity of enzymes related to nitrogen (N) metabolism of crop plants could increase with the increase in soil nitrogen content [15]. Gong et al. [5] found that with intercropping between millet and mung bean, the activity of soil enzymes, i.e., invertase (INV), urease (URE), and catalase (CAT), increased significantly by 18.39%, 29.09%, and 42.90%, respectively. Soil alkaline protease (S-ALPT) acts on the mineralization of organic nitrogen and decomposes proteins and peptides in plant residues in the soil into amino acids [16], so the activity of soil protease directly affects the intensity of nitrogen transformation in the soil and the nitrogen supply capacity of the soil [17]. More than 95% of $NH_4^+$ is assimilated by glutamine oxoglutarate aminotransferase (GOGAT)/glutamate synthase (GS) in plants [18,19]. No matter the nitrogen absorbed by plants or the ammonia released by metabolism, it needs GS catalysis to assimilate into amino acids and then synthesize proteins [20]. Nitrate reductase (NR) is involved in nitrogen assimilation in plants, and NR degrades $NO^{3-}$ to $NO^{2-}$; it plays an important role in the process of plant nitrogen absorption [21].

At present, the excessive application of chemical fertilizers, especially nitrogen fertilizer, is a serious problem in agricultural production. The low efficiency of crop fertilizer absorption or utilization not only causes a large amount of fertilizer waste, but also brings serious farmland environmental pollution [22]. The excessive application of nitrogen fertilizer did not affect land equivalent ratio (LER), but decreased the grain yield [13]. A more efficient, green, and sustainable agricultural production model urgently needs to be developed and applied for crop production. Intercropping is a suitable method to stabilize the yield in the production system with low nitrogen input [23]. Scientific and rational fertilization, to a certain extent, can reduce the use of nitrogen fertilizer, simultaneously improving nitrogen use efficiency (NUE) and increasing crop production [24,25]. Legumes, e.g., soybean and peanut, called nitrogen-fixing capacity (NFC) crops, can fix nitrogen through rhizobia to increase soil nitrogen content. The legume-grain intercropping can increase soil fertility and reduce soil nitrogen loss [22,26]. Some studies have shown that higher recovery efficiency can be achieved when nitrogen supply is reduced by 100 kg/ha without reducing summer corn or winter wheat crop yields [27,28]. A meta-analysis of a large literature found that intercropping can increase yields by 16–29% per hectare; compared with a sole crop, the amount of chemical fertilizer was reduced by 19–35% [29].

Intercropping can increase crop yield [30,31], usually producing a higher yield per unit of land than a sole crop [12], e.g., wheat-corn intercropping can increase overall grain yield [32]. The intercropping of cereal crops with legumes can increase land equivalent, such as corn-pigeonpea intercropping and corn-peanut intercropping [13,33,34]. Soybean, as a high NFC crop, can obtain a large amount of nitrogen from soil through its rhizobia resource. Some studies have shown that under sufficient irrigation conditions, the root system of corn-soybean intercropping ecosystem was not only deeper than soybean, but also extended to the soybean part of the field to absorb more resources [35]. The intercropping of peanut with *Atractylodes Lancea* effectively controlled peanut disease and increased yield [36].

The soil microbial community plays an important role in regulating soil ecosystem species [37]. Soil fungi play an important role in the nutrient cycle and biological interaction and can provide mineral nutrients to plants [38]. Chemical fertilizers can affect the activity of soil microorganisms [39,40] and also the composition of individual microbial

communities in the soil [41]. In contrast, the NFC crops can increase the amount of nitrogen in the soil, which can affect the microbial community in the soil [42,43].

Corn, peanut, soybean, and millet are the main food or oil crops in China, and they are all summer crops and have the same growing season, which makes them suitable for intercropping cultivation. The purpose of this study was to explore a crop suitable for intercropping with corn in North China and screen out the best intercropping type and planting pattern.

## 2. Materials and Methods

### 2.1. Experimental Location

The experiment was conducted in Nanxin Village, Jiyang County of Shandong Province of China (36°58′ N, 117°13′ E), located in the warm temperate and semi-humid monsoon climate zone, with annual average values of temperature of 12.8 °C, frost-free period of 195 days, solar radiation of 124.4 kcal/cm$^2$, and precipitation of 583.3 mm. The precipitation is mostly concentrated between the jointing and dough stages of corn crop. The land of Jiyang County belongs to the plain area, the soil layer is 30–40 cm deep, and the soil type is fluvo-aquic soil. (The soil formed by river sediment affected by groundwater movement and farming activities. It is named after the phenomenon of night tide.) The experimental site has a total of 4.5 hectares and is rectangular, with a length of 300 m from north to south and a width of 154 m from east to west.

### 2.2. Crop Cultivars

Two types of grain crops were used in this study, including the non-NFC crops, i.e., corn (cv. Liangyu 99; provided by Dandong Denghai Liangyu Seed Industry Co., Ltd. from Dandong City, Liaoning Province of China) and millet (cv. Lazy Valley No. 3; provided by the Lutong Seed Industry Co., Ltd. from Yongnian County, Hebei Province, China); and the NFC crops, i.e., soybean (cv. Xindou 1; provided by Jinan Zhaohui Seed Industry Co., Ltd. from Jinan City of Shandong Province of China) and peanut (cv. Huayu 22; provided by the Shandong Peanut Research Institute at Qingdao City of Shandong Province of China). These four selected crops are all summer-autumn crops and have the same growing season.

### 2.3. Experiment Setup

In this experiment, the replacement intercropping [44] was used. There were three intercropping types (i.e., corn-soybean/peanut/millet intercropping), five planting patterns (including three planting patterns of corn intercropped with the NFC crop (soybean or peanut), and with the non-NFC crop (millet) as two corn rows to 2, 3, and 4 rows of soybean/peanut and to 2, 4, and 6 rows of millet, and two sole crop patterns of corn and soybean/peanut/millet), and two fertilizer levels (i.e., normal (600 kg/ha) and reduced (375 kg/ha) levels of NPK (N:P$_2$O$_5$:K$_2$O = 15:15:15) fertilizer), and total 3 intercropping types (T) × 5 planting patterns (P') × 2 fertilizer levels (F) = 30 treatments. There were 3 plots for each treatment, and a total of 90 plots in this experiment. In addition, the row spacing and hill spacing of corn, soybean, and peanut were all 0.80 m and 0.20 m respectively, and those of millet were 0.20 m and 0.05 m respectively, and the spacing of two millet rows to neighboring two millet rows was 0.80 m. The spacing of a corn row to the row of soybean, peanut, and millet was 0.80 m, respectively (Figure 1); corn and millet each hole one plant, peanut and soybean each hole two plants. The planting density of corn was approximately 6.25 plants/m$^2$, soybean and peanut were 12.5 plants/m$^2$, and millet was 40 plants/m$^2$. Moreover, the length and width of each plot were 28.8 m and 15.5 m, respectively, and there was 1.0 m spacing between neighboring plots.

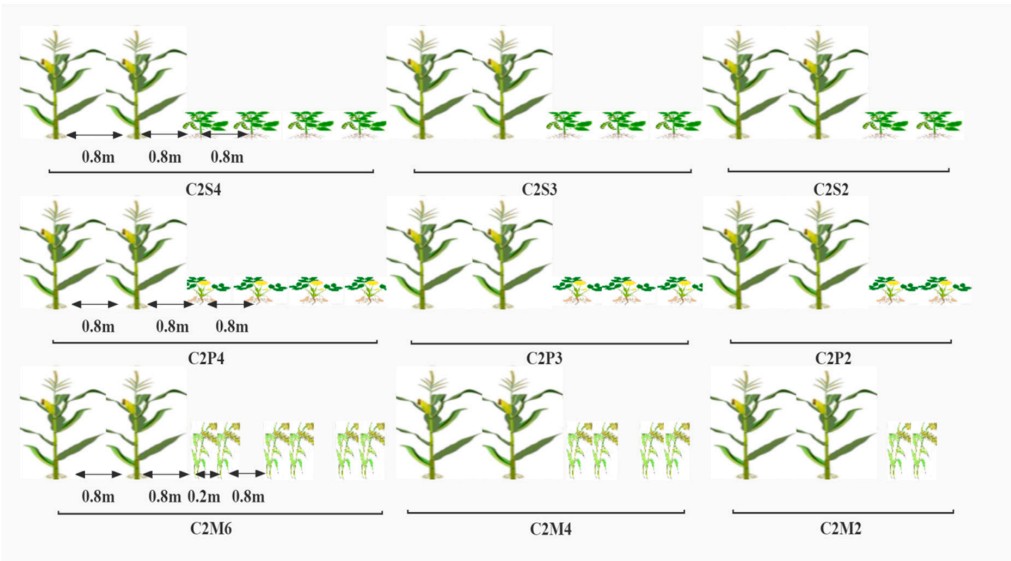

**Figure 1.** Diagram of different intercropping types (i.e., corn-soybean/peanut/millet intercropping) and planting patterns (including three intercropping patterns of corn with soybean, peanut, and millet) in this study. Soybean (S), peanut (P), millet (M), and corn (C); C2S2, C2S3, C2S4—Intercropping of 2 corn rows with 2, 3, 4 rows of soybean, respectively; C2P2, C2P3, C2P4—Intercropping of 2 corn rows with 2, 3, 4 rows of peanut, respectively; C2M2, C2M4, C2M6—Intercropping of 2 corn rows with 2, 4, 6 rows of millet, respectively. The row spacing and hill distance of corn, soybean, and peanut were all 0.80 m and 0.20 m, and those of millet were 0.80 m and 0.05 m, respectively, and the spacing of 2 rows to neighboring 2 rows of millet was 0.80 m. The spacing of a corn row to the row of soybean, peanut, and millet was 0.80 m.

### 2.4. Sample Collection and Determination

All the crops of corn (C), soybean (S), peanut (P), and millet (M) were sown on June 16 of 2018 and 2019, and the planting cultivation adopted was the same field management measures in the local fields, including s one-time spraying of herbicides before the planting of corn (Lasso provided by the Monsanto Company), soybean (Clethodim provided by the Chevron Phillips Chemical Company LP), peanut (Fluazifop-butyl provided by the Shandong Luba Chemical Co., Ltd., Jinan, China), and millet (Acetamide herbicide provided by the Tianjin Herite Biological Technology Co., Ltd., Tianjin, China) and a one-time irrigation before planting. No insecticides were sprayed during the entire growing seasons of 2018 and 2019.

On July 25 (jointing stage) and September 10 (dough stage) in 2018 and 2019, 10 plants of corn were randomly selected from each plot, and the upper first leaf of corn was collected for the following plant assay. At the same time, the rhizosphere soil of the sampled plants was also collected for the following soil assay. All the collected plants and soil samples were stored in dry ice and brought to the laboratory immediately for the detection of the activity of N-metabolism-related enzymes in corn leaves (including glutamine oxoglutarate aminotransferase (GOGAT), glutamate synthetase (GS), nitrate reductase (NR)) and the activity of N-metabolism-related enzymes in corn rhizosphere soil (i.e., soil alkaline protease (S-ALPT), which were assayed by using the reagent kits, BC0075 (GOGAT-activity Test, micromethod), BC0915 (GS-activity Test, micromethod), BC0085 (NR-activity Test, micromethod) and BC0885 (S-ALPT-activity Test, micromethod)). All the kits were provided by the Beijing Solarbio Science & Technology Co., Ltd., Beijing, China. During the corn harvest period, 10 adjacent corn plants were randomly selected from each plot, 3 replicates per plot, dried in the sun to constant weight, and then the biomass and grain yield of each plant were measured by electronic balance (accuracy: 0.1 g; range: 0–5 kg; Shanghai Yaohua Weighing System Co., Ltd., Shanghai, China). A thousand grains were randomly selected from 10 corn plants in each group and weighed with 1000-grain weight,

three replicates per plot. According to the formula: yield per hectare (kg) = number of plants at sample point/area of sample point (m$^2$) × 10,000 × grain yield per plant (g)/1000, the grain yield per hectare is obtained. At the jointing stage, the total nitrogen content in the collected leaves and soil was determined by the Kjeldahl method.

Based on the effects of intercropping with soybean, peanut, and millet on the biomass and grain yield of corn, the best intercropping patterns (including C2S4-C, C2P4-C, and C2M6-C with the sole crop pattern of corn (C) as the control) under normal and reduced fertilizer were selected from all the three intercropping types, and the microbial community of corn rhizosphere soil was analyzed in 2019. Total genomic DNA samples were extracted using the OMEGA Soil DNA Kit (M5635-02) (Omega Bio-Tek, Norcross, GA, USA), following the manufacturer's instructions, and stored at −20 °C prior to further analysis. The quantity and quality of extracted DNAs were measured using a NanoDrop NC2000 spectrophotometer (Thermo Fisher Scientific, Waltham, MA, USA) and agarose gel electrophoresis, respectively. PCR amplification of the bacterial 16S rRNA gene V3–V4 region was performed using the forward primer 338F (5′-ACTCCTACGGGAGGCAGCA-3′) and the reverse primer 806R (5′-GGACTACHVGGGTWTCTAAT-3′). Sample-specific 7-bp barcodes were incorporated into the primers for multiplex sequencing. The PCR components contained 5 μL of buffer (5 × ), 0.25 μL of Fast pfu DNA Polymerase (5U/μL), 2 μL (2.5 mM) of dNTPs, 1 μL (10 μM) of each forward and reverse primer, 1 μL of DNA template, and 14.75 μL of ddH$_2$O. Thermal cycling consisted of initial denaturation at 98 °C for 5 min, followed by 25 cycles of denaturation at 98 °C for 30 s, annealing at 53 °C for 30 s, and extension at 72 °C for 45 s, with a final extension of 5 min at 72 °C. PCR amplicons were purified with Vazyme V AHTSTM DNA Clean Beads (V azyme, Nanjing, China) and quantified using the Quant-iT PicoGreen dsDNA Assay Kit (Invitrogen, Carlsbad, CA, USA). After the individual quantification step, amplicons were pooled in equal amounts, and pair-end 2250 bp sequencing was performed using the Illlumina MiSeq platform with MiSeq Reagent Kit v3 at Shanghai Personal Biotechnology Co., Ltd. (Shanghai, China).

This project used the Illumina platform to perform paired-end sequencing of community DNA fragments. We used DADA2 [45] for sequence denoising: qiime cutadapt trim-paired was used to remove the sequence of primer fragments, the unmatched primer sequence was discarded; then qiime dada2 denoise-paired was used to call DADA2 for quality control, denoising, splicing, and de-chimerism. The above steps were analyzed separately for each library. After denoising all libraries, we merged the ASVs feature sequence and the advance super view (ASV) table and removed the singleton ASVs. The R language script was used to create statistics on the length distribution of the high-quality sequences contained in all samples.

Species taxonomy notes: (1) For the 16S rRNA genes of bacteria or archaea, the Greengenes database was used (Release 13.8, http://greengenes.secondgenome.com/, accessed on 16 October 2019) [46]; (2) For the 18S rRNA gene of eukaryotic microorganisms, the nt database was used (ftp://ftp.ncbi.nih.gov/blast/db/, accessed on 18 October 2019); (3) For fungal ITS sequences, the UNITE database was used (Release 8.0, https://unite.ut.ee/, accessed on 18 October 2019) [47]; (4) For functional genes or other requirements, the nt or nr databases were used for annotation (ftp://ftp.ncbi.nih.gov/blast/db/, accessed on 18 October 2019). Analysis steps: (1) For the first three types of databases, we used QIIME2's classify-sklearn algorithm [48] (https://github.com/QIIME2/q2-feature-classifier, accessed on 20 October 2019). For each ASV feature sequence or each A representative sequence of an OTU, we used the default parameters in the QIIME2 software and the pre-trained Naive Bayes classifier for species annotation. (2) For the nt or nr database, we used the BROCC algorithm [49] (https://github.com/kylebittinger/q2-brocc#the-brocc-algorithm, accessed on 6 November 2019). We used blastn or balstx to compare the sequence with Align, the nucleic acid or protein sequence in the nt or nr database; we then called the brocc.py script to obtain the annotation information according to the recommended parameters. It should be noted that, according to the principle of database selection described above, the nt or nr database here generally does not refer to the entire database but

is limited according to the accession number or Gi number of the sequencing target species and/or gene name Database subset.

Sequence denoising or operational taxonomic units (OTUs) clustering was performed according to the QIIME2 DADA2 [45] analysis process or the Vsearch [50] software analysis process. The specific composition of each sample (group) at different taxonomic levels was displayed to understand the overall situation. According to the distribution of amplicon sequence variants (ASV)/OTU [51,52] in different samples, the alpha diversity level of each sample was evaluated to reflect whether the sequencing depth was appropriate through a sparse curve. In order to comprehensively assess the alpha diversity of the microbial community, Observed Species indices were used in this process to characterize the richness. Shannon [53] and Simpson [54] indices were used to characterize diversity. At the ASV/OTU level, the distance matrix of each sample was calculated, and the differences in beta diversity and significance among different samples (groups) were measured by a variety of unsupervised sorting and clustering methods combined with corresponding statistical test methods. At the level of taxonomic composition, the diversity of species abundance composition among different samples (groups) was further measured through various unsupervised sorting, clustering, and modeling methods, combined with corresponding statistical test methods, in order to find marker species. According to the species composition and distribution in each sample, the association network was constructed to calculate the topological index, in order to find the key species. Based on the sequencing results of 16S rRNA, 18S rRNA, and internal transcribed spacer (ITS) genes, the metabolic function of samples was predicted to discover the differential pathways and obtain the species composition of specific pathways (https://github.com/picrust/picrust2/wiki, accessed on 6 November 2019). Analysis software used for this analysis included QIIME2 (2019.4), R language, and GGploT2 package. On 10 October 2018 and 2019, five points were randomly selected from each plot using a five-point sampling method, five corn plants were harvested from each sampling point, and the sampled corn plants were sun-dried in the field for one week to weigh the dry weight using the same electronic balance (Shanghai Yaohua weighing system Co., Ltd., Shanghai, China).

*2.5. Data Statistical Analysis*

All data were analyzed with SPSS 20.0. Four-way ANOVA was used to analyze the effects of sampling year (Y), fertilizer level (F), intercropping type (T), planting pattern (P'), and their interactions on the biomass per plant, 1000-grain weight, yield per hectare, and the activity of N-metabolism-related enzymes. In addition, two-way ANOVA was used to analyze the effects of fertilizer level (F) and planting pattern (P') and their interactions on the soil microbial composition. Significant differences between or among treatments were analyzed by using LSD test at $p < 0.05$.

## 3. Results

*3.1. Corn Growth and Production of Corn in Intercropping System*

Sampling year (Y), fertilizer level (F), intercropping type (T), and planting pattern (P') significantly affected the biomass per plant of corn ($p < 0.05$), as shown in Table 1. However, fertilizer level had no significant effect on corn biomass in the dough stage ($p = 0.356$). In addition, the interaction between intercropping type and planting pattern had a significant effect on corn biomass in the jointing stage ($p < 0.001$), and the interaction between fertilizer level and planting pattern had a significant effect on corn biomass in the dough stage ($p = 0.003$). Sampling year ($p < 0.001$), fertilizer level ($p = 0.011$), intercropping type ($p < 0.001$), planting pattern ($p < 0.001$), and the interaction between intercropping type and planting pattern ($p = 0.004$) all had significant effects on ear weight per plant of corn. Sampling year ($p = 0.045$) and intercropping type ($p = 0.003$) had significant effects on 1000-grain weight. Sampling year ($p < 0.001$), intercropping type ($p < 0.001$), planting pattern ($p < 0.001$), and the interaction between intercropping type and planting pattern ($p = 0.005$) all had significant effects on the grain weight per plant. Sampling year

($p < 0.001$), intercropping type ($p < 0.001$), planting pattern ($p < 0.001$), and the interaction between intercropping type and planting pattern ($p = 0.003$) all had significant effects on yield.

**Table 1.** Four-factor variance analysis of sampling year (2018 vs. 2019), fertilizer level (normal vs. reduced), intercropping type (corn-soybean/peanut/millet), and planting pattern (three patterns of intercropping of corn with soybean/peanut/millet, and one sole crop pattern of corn) on the biomass per plant, 1000-grain weight, and grain yield per hectare of corn plants (*F/p* value).

| Source of Variation | Biomass per Plant (g) | | 1000-Grain Weight (g) | | Grain Yield (kg/ha) | |
|---|---|---|---|---|---|---|
| | *F* | *p* | *F* | *p* | *F* | *p* |
| Sampling year (Y) | 7.125 | 0.008 ** | 4.047 | 0.045 * | 57.273 | <0.001 *** |
| Fertilizer (F) | 0.855 | 0.356 | 0.105 | 0.746 | 3.111 | 0.080 |
| Intercropping type (T) | 5.378 | 0.005 ** | 6.057 | 0.003 ** | 98.454 | <0.001 *** |
| Planting pattern (P′) | 15.154 | <0.001 *** | 0.874 | 0.418 | 11.661 | <0.001 *** |
| Y × F | 0.597 | 0.441 | 0.046 | 0.830 | 12.992 | <0.001 *** |
| Y × T | 2.793 | 0.064 | 2.769 | 0.065 | 3.507 | 0.032 * |
| Y × P′ | 6.319 | <0.001 *** | 1.764 | 0.173 | 3.642 | 0.028 * |
| F × T | 0.788 | 0.456 | 0.122 | 0.885 | 0.119 | 0.888 |
| F × P′ | 3.611 | 0.014 * | 0.286 | 0.751 | 0.177 | 0.838 |
| T × P′ | 1.955 | 0.074 | 0.856 | 0.491 | 4.264 | 0.003 ** |
| Y × F × T | 2.166 | 0.117 | 0.156 | 0.856 | 0.851 | 0.429 |
| Y × F × P′ | 4.855 | 0.003 ** | 0.169 | 0.845 | 0.456 | 0.634 |
| Y × T × P′ | 1.995 | 0.068 | 0.192 | 0.942 | 1.224 | 0.303 |
| F × T × P′ | 1.909 | 0.081 | 0.610 | 0.656 | 0.522 | 0.720 |
| Y × F × T × P′ | 1.559 | 0.161 | 0.722 | 0.578 | 0.118 | 0.976 |

\* $p < 0.05$; \*\* $p < 0.01$; \*\*\* $p < 0.001$.

### 3.1.1. Biomass per Plant of Corn

Intercropping significantly increased the biomass per plant of corn. The biomass per plant of corn under corn-peanut intercropping was significantly increased compared with corn sole crop. The data showed that the biomass per plant of corn under corn-peanut intercropping had an increasing but insignificant trend. There was no significant difference between corn-millet intercropping and corn sole crop ($p < 0.05$; Table 1). At the jointing stage, corn biomass per plant increased by 15.32% (2018, Figure 2a) and 35.15% (2019, Figure 2b) under normal fertilizer compared with corn sole crop on average. Compared with corn sole crop, corn biomass per plant increased by 24.63% (2018, Figure 2a) and 26.40% (2019, Figure 2b) on average in corn-peanut intercropping with reduced fertilizer application. At the dough stage, corn biomass per plant increased by 11.44% (2018, Figure 2a) and 4.87% (2019, Figure 2b) on average under normal fertilizer compared with corn sole crop. Corn-peanut intercropping with reduced fertilizer application increased corn biomass per plant by 8.65% (2018, Figure 2a) and 9.16% (2019, Figure 2b) on average. In general, there was no significant difference among different planting patterns under the same intercropping type and fertilizer. In addition, fertilizer had no significant effect on corn biomass in different periods under the same planting pattern as a whole.

### 3.1.2. 1000-Grain Weight of Corn

Intercropping type, planting pattern, and fertilizer had no significant effect on 1000-grain weight of corn in the first year, but intercropping type and planting pattern had significant effect on it in the second year ($p < 0.05$, Table 1). The 1000-grain weight of corn under normal fertilizer increased by 4.85–13.28% (corn-soybean, Figure 3b), 9.16–15.46% (corn-peanut, Figure 3b), and 5.62–9.94% (corn-millet, Figure 3b), respectively, compared with the sole crop. Under reduced fertilizer application, the 1000-grain weight of corn increased by 6.37–19.22% (corn-soybean, Figure 3b), 18.91–21.56% (corn-peanut, Figure 3b), and 11.57–14.27% (corn-millet, Figure 3b), respectively.

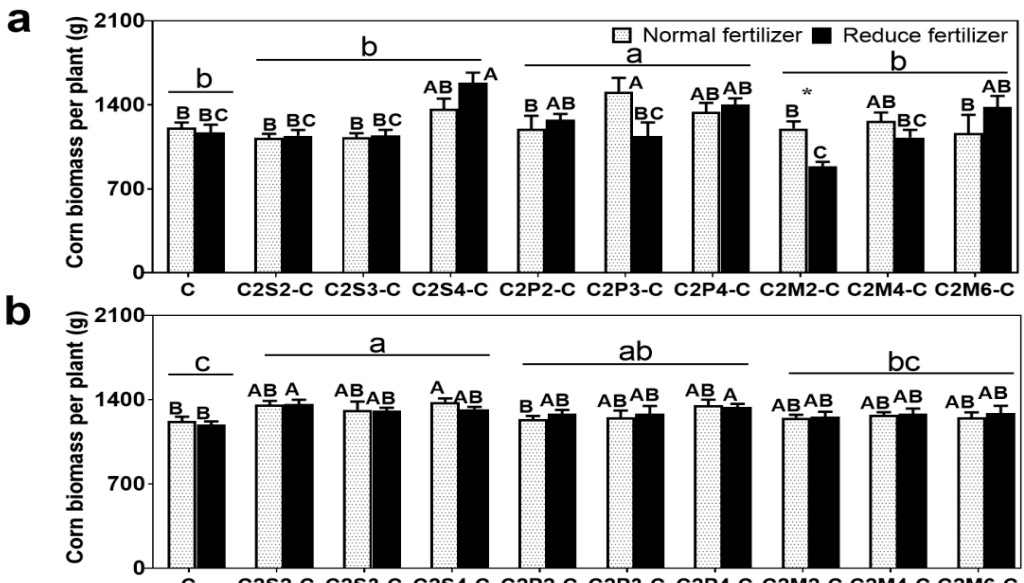

**Figure 2.** Corn biomass per plant under different planting patterns in 2018 (**a**) and 2019 (**b**). Different lowercase and uppercase letters represented significant differences among different intercropping types and the corn sole crop, and among different planting patterns at the same fertilizer level by LSD test at $p < 0.05$; *—significant differences between the normal and reduced fertilizer levels in the same planting pattern. C—corn, S—soybean, P—peanut, M—millet; C2S2, C2S3, C2S4—intercropping of 2 corn rows with 2, 3, 4 rows of soybean, respectively; C2P2, C2P3, C2P4—intercropping of 2 corn rows with 2, 3, 4 rows of peanut, respectively; C2M2, C2M4, C2M6—intercropping of 2 corn rows with 2, 4, 6 rows of millet, respectively.

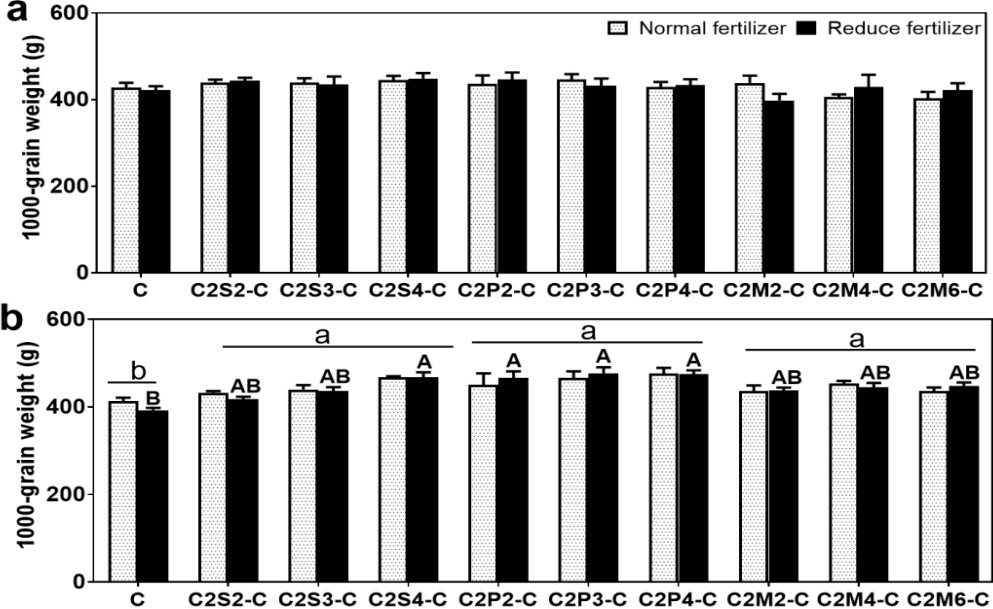

**Figure 3.** The 1000-grain weight of corn under different planting patterns in 2018 (**a**) and 2019 (**b**). Different lowercase and uppercase letters represented significant differences among different intercropping types and the corn sole crop, and among different planting patterns at the same fertilizer level by LSD test at $p < 0.05$. C—corn, S—soybean, P—peanut, M—millet; C2S2, C2S3, C2S4—intercropping of 2 corn rows with 2, 3, 4 rows of soybean, respectively; C2P2, C2P3, C2P4—intercropping of 2 corn rows with 2, 3, 4 rows of peanut, respectively; C2M2, C2M4, C2M6—intercropping of 2 corn rows with 2, 4, 6 rows of millet, respectively.

### 3.1.3. Grain Yield of Corn per Hectare

Corn grain yield under corn-soybean and corn-peanut intercropping was significantly higher than that under corn sole crop and corn-millet intercropping, and data in the second year showed that corn-soybean intercropping was significantly higher than that under corn-peanut intercropping ($p < 0.05$, Table 1). Compared with the corn sole crop, corn-soybean and corn-peanut intercropping significantly increased corn grain yield. Under normal fertilizer, corn grain yield increased by 12.18–17.47% (corn-soybean, 2018, Figure 4a) and 9.89–11.72% (corn-peanut, 2018, Figure 4a), respectively, compared with the corn sole crop. Under reduced fertilizer application, corn grain yield increased by 16.29–20.15% (corn-soybean, 2018, Figure 4a) and 13.18–19.65% (corn-peanut, 2018, Figure 4a), respectively. Under normal fertilizer, corn grain yield increased by 11.31–31.21% (corn-soybean, 2019, Figure 4b) and 5.72–16.78% (corn-peanut, 2019, Figure 4b), respectively, compared with the corn sole crop. Under reduced fertilizer application, corn grain yield increased by 14.36–27.44% (corn-soybean, 2019, Figure 4b) and 5.39–16.67% (corn-peanut, 2019, Figure 4b), respectively. In addition, it can be seen from the data that the grain yield increase effect of corn-soybean intercropping was stronger than that of corn-peanut intercropping, which reached a significant level in 2019 (Figure 4b). There were significant regular differences in corn grain yield in planting patterns, especially in the corn-soybean and corn-peanut intercropping patterns. As the proportion of corn in row decreased, the grain yield of corn increased (Figure 4a,b).

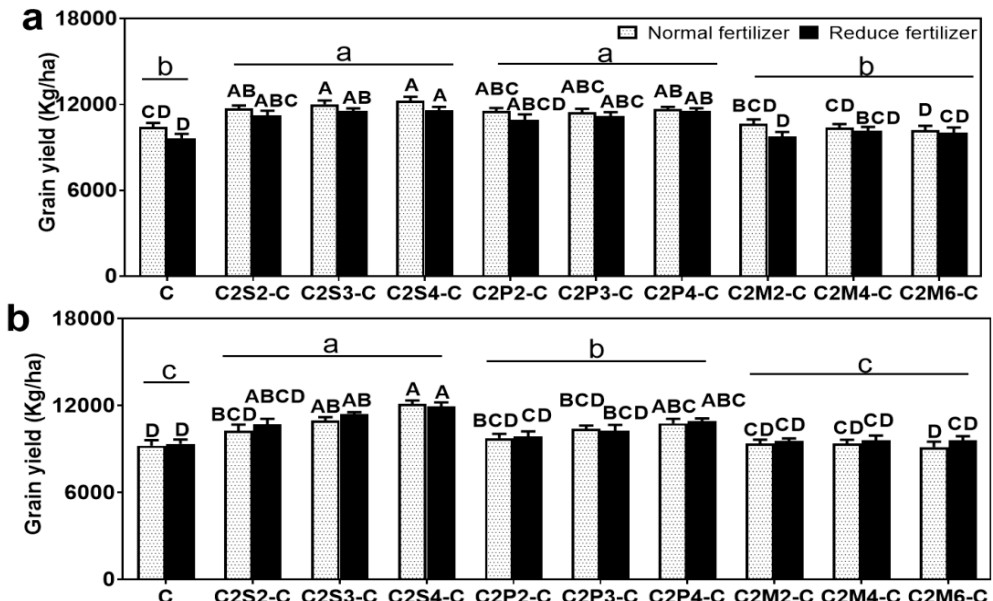

**Figure 4.** Grain yield per hectare of corn under different planting patterns in 2018 (**a**) and 2019 (**b**). Different lowercase and uppercase letters represented significant differences among different intercropping types and the corn sole crop, and among different planting patterns at the same fertilizer level by LSD test at $p < 0.05$. C—corn, S—soybean, P—peanut, M—millet; C2S2, C2S3, C2S4—intercropping of 2 corn rows with 2, 3, 4 rows of soybean, respectively; C2P2, C2P3, C2P4—intercropping of 2 corn rows with 2, 3, 4 rows of peanut, respectively; C2M2, C2M4, C2M6—intercropping of 2 corn rows with 2, 4, 6 rows of millet, respectively.

### 3.2. Activity of Nitrogen (N) Metabolism-Related Enzymes

The activity of individual enzymes varied in different years. Examples include S-ALPT ($p = 0.045$, Table 2) and NR ($p = 0.049$) at the jointing stage and GOGAT ($p = 0.005$), GS ($p < 0.001$), and NR ($p < 0.001$) at the dough stage. Fertilizer had no significant effect on enzyme activity in different periods ($p > 0.05$). There were significant differences in enzyme activities in different intercropping types in different periods ($p < 0.001$). Except for S-ALPT

at the jointing stage, planting patterns had significant or extremely significant effects on enzyme activities at all stages ($p < 0.05$). Under the interaction, fertilizer and intercropping type ($p = 0.016$) and intercropping type and planting pattern ($p < 0.001$) had significant or extremely significant effects on S-ALPT activity at the jointing stage. In addition, the interaction of intercropping type and planting pattern had significant effects on S-ALPT ($p < 0.001$) and GOGAT ($p = 0.008$) activities at the dough stage.

**Table 2.** Four-factor variance analysis of nitrogen metabolization-related enzymes in corn under different sampling years, fertilizer, intercropping types, and planting patterns at the jointing and dough stages ($F/p$ value).

| Source of Variation | Jointing Stage | | | | | | | | Dough Stage | | | | | | | |
| --- | --- | --- | --- | --- | --- | --- | --- | --- | --- | --- | --- | --- | --- | --- | --- | --- |
| | S-ALPT | | GOGAT | | GS | | NR | | S-ALPT | | GOGAT | | GS | | NR | |
| | F | p | F | p | F | p | F | p | F | p | F | p | F | p | F | p |
| Sampling year (Y) | 4.141 | 0.045 * | 0.101 | 0.751 | 0.739 | 0.392 | 4.012 | 0.049 * | 0.213 | 0.646 | 8.350 | 0.005 ** | 79.506 | <0.001 *** | 107.677 | <0.001 *** |
| Fertilizer (F) | 2.651 | 0.107 | 0.508 | 0.478 | 0.126 | 0.723 | 1.629 | 0.206 | 1.191 | 0.278 | 0.069 | 0.793 | 0.614 | 0.435 | 0.000 | 1.000 |
| Intercropping type (T) | 248.395 | <0.001 *** | 173.555 | <0.001 *** | 222.866 | <0.001 *** | 299.996 | <0.001 *** | 151.385 | <0.001 *** | 92.373 | <0.001 *** | 306.094 | <0.001 *** | 161.850 | <0.001 *** |
| Planting pattern (P′) | 1.900 | 0.156 | 4.661 | 0.012 * | 6.923 | 0.002 ** | 7.321 | 0.0012 * | 7.824 | <0.001 *** | 4.664 | 0.012 * | 40.716 | <0.001 *** | 22.260 | <0.001 *** |
| Y × F | 0.802 | 0.373 | 0.088 | 0.768 | 0.274 | 0.602 | 1.944 | 0.167 | 0.617 | 0.434 | 0.066 | 0.798 | 0.475 | 0.493 | 0.001 | 0.974 |
| Y × T | 0.662 | 0.519 | 2.915 | 0.060 | 1.760 | 0.179 | 1.131 | 0.328 | 0.078 | 0.925 | 1.532 | 0.222 | 1.818 | 0.169 | 8.016 | <0.001 *** |
| Y × P′ | 1.474 | 0.235 | 3.289 | 0.042 * | 1.844 | 0.165 | 0.404 | 0.669 | 1.363 | 0.262 | 1.096 | 0.339 | 0.221 | 0.802 | 1.203 | 0.306 |
| F × T | 0.651 | 0.525 | 0.560 | 0.573 | 0.032 | 0.968 | 1.123 | 0.331 | 0.618 | 0.542 | 1.608 | 0.207 | 1.765 | 0.178 | 0.001 | 0.999 |
| F × P′ | 4.373 | 0.016 * | 1.410 | 0.250 | 0.355 | 0.702 | 0.066 | 0.936 | 0.937 | 0.396 | 0.514 | 0.600 | 0.137 | 0.872 | 0.009 | 0.991 |
| T × P′ | 5.386 | <0.001 *** | 1.217 | 0.310 | 3.645 | 0.009 ** | 1.380 | 0.248 | 5.316 | <0.001 *** | 3.738 | 0.008 ** | 1.453 | 0.224 | 1.660 | 0.167 |
| Y × F × T | 1.201 | 0.306 | 0.224 | 0.800 | 0.211 | 0.810 | 0.270 | 0.764 | 0.446 | 0.641 | 0.101 | 0.904 | 0.183 | 0.833 | 0.001 | 0.999 |
| Y × F × P′ | 1.092 | 0.341 | 0.118 | 0.889 | 0.161 | 0.851 | 0.603 | 0.550 | 0.150 | 0.861 | 0.011 | 0.989 | 0.101 | 0.904 | 0.115 | 0.892 |
| Y × T × P′ | 2.331 | 0.063 | 1.208 | 0.314 | 0.619 | 0.650 | 0.777 | 0.543 | 0.546 | 0.703 | 0.170 | 0.953 | 0.372 | 0.828 | 0.683 | 0.606 |
| F × T × P′ | 2.365 | 0.060 | 2.468 | 0.051 | 1.322 | 0.269 | 0.480 | 0.750 | 0.765 | 0.551 | 0.211 | 0.932 | 0.434 | 0.784 | 0.906 | 0.464 |
| Y × F × T × P′ | 1.717 | 0.154 | 0.465 | 0.761 | 0.271 | 0.896 | 0.516 | 0.724 | 0.514 | 0.726 | 0.048 | 0.996 | 0.168 | 0.954 | 0.160 | 0.958 |

S-ALPT—soil alkaline protease; GOGAT—glutamine oxoglutarate aminotransferase; GS—glutamate synthetase; NR—nitrate reductase; * $p < 0.05$; ** $p < 0.01$; *** $p < 0.001$.

For S-ALPT, the enzyme activity in corn-soybean and corn-peanut intercropping was significantly higher than that in the corn sole crop, and the enzyme activity in corn-soybean intercropping was significantly higher than that in corn-peanut intercropping. At the corn jointing stage, S-ALPT activity increased by 67.36–110.75% (corn-soybean, 2018, Figure 5a), 73.08–103.21% (corn-soybean, 2019, Figure 5c), 46.46–53.88% (corn-peanut, 2018, Figure 5a), and 44.77–53.42% (corn-peanut, 2019, Figure 5c) significantly under normal fertilizer compared with the corn sole crop, respectively. Under reduced fertilizer application, S-ALPT activity significantly increased by 101.49–132.51% (corn-soybean, 2018, Figure 5a), 95.26–124.84% (corn-soybean, 2019, Figure 5c), 87.95–99.50% (corn-peanut, 2018, Figure 5a), and 74.67–100.65% (corn-peanut, 2019, Figure 5c). At the dough stage, S-ALPT activity significantly increased by 40.72–81.80% (corn-soybean, 2018, Figure 5b), 45.21–74.90% (corn-soybean, 2019, Figure 5d), 8.65–28.83% (corn-peanut, 2018, Figure 5d), and 15.52–36.97% (corn-peanut, 2019, Figure 5d) under normal fertilizer compared with the corn sole crop, respectively. Under reduced fertilizer application, S-ALPT activity increased significantly by 52.06–107.00% (corn-soybean, 2018, Figure 5b), 59.66–79.73% (corn-soybean, 2019, Figure 5d), 29.84–32.72% (corn-peanut, 2018, Figure 5b), and 15.91–24.62% (corn-peanut, 2019, Figure 5d).

For GOGAT, the enzyme activity in corn-soybean and corn-peanut intercropping was significantly higher than that in the corn sole crop, and the enzyme activity in corn-soybean intercropping was significantly higher than that in corn-peanut intercropping. At the corn jointing stage, intercropping with normal fertilizer significantly increased GOGAT activity by 32.14–56.25% (corn-soybean, 2018, Figure 6a), 40.49–59.87% (corn-soybean, 2019, Figure 6c), 16.45–18.76% (corn-peanut, 2018, Figure 6a), and 18.89–38.39% (corn-peanut, 2019, Figure 6c), respectively, compared with the corn sole crop. Under reduced fertilizer application, S-ALPT activity significantly increased by 35.21–39.49% (corn-soybean, 2018, Figure 6a), 40.70–47.60% (corn-soybean, 2019, Figure 6c), 9.10–19.01% (corn-peanut, 2018, Figure 6a), and 23.11–34.91% (corn-peanut, 2019, Figure 6c). At the dough stage, the S-ALPT activity under normal fertilizer significantly increased by 11.66–22.43% (corn-soybean, 2018, Figure 6b), 18.82–35.41% (corn-soybean, 2019, Figure 6d), 4.96–8.40% (corn-peanut, 2018, Figure 6b), and 15.65–16.99% (corn-peanut, 2019, Figure 6d). Under reduced fertilizer

application, the activity of S-ALPT significantly increased by 19.97–29.30% (corn-soybean, 2018, Figure 6b), 32.82–45.13% (corn-soybean, 2019, Figure 6d), 8.57–13.00% (corn-peanut, 2018, Figure 6b), and 22.64–27.17% (corn-peanut, 2019, Figure 6d).

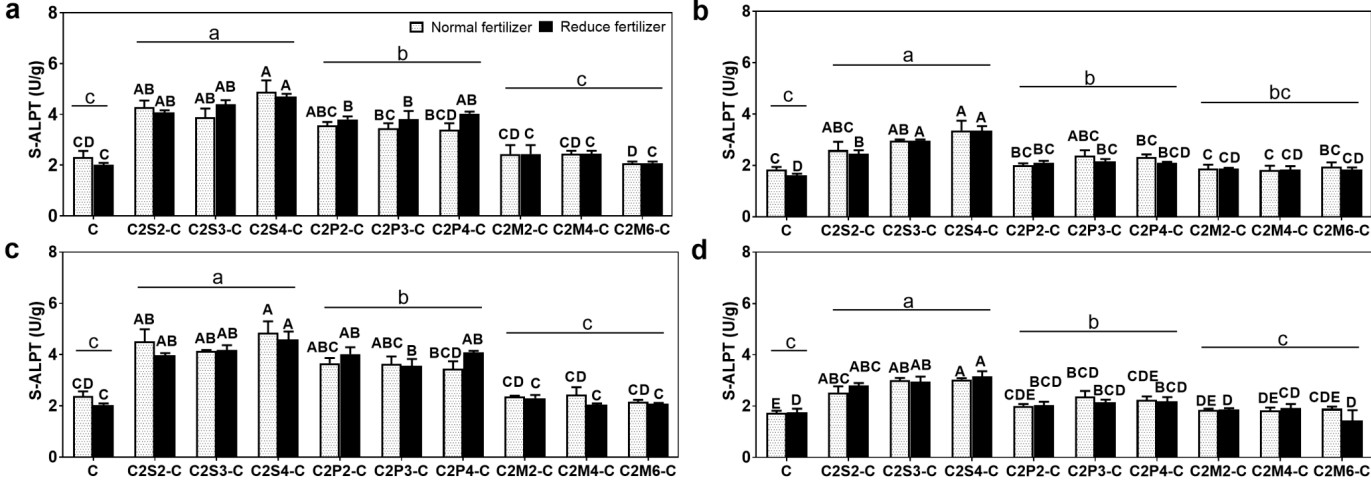

**Figure 5.** S-ALPT enzyme activity of corn under different planting patterns in 2018 (**a**,**b**) and 2019 (**c**,**d**): (**a**,**c**)—jointing stage; (**b**,**d**)—dough stage; lowercase letters represent differences between intercropping types; uppercase letters represent differences between planting patterns at the same fertilizer level. C—corn, S—soybean, P—peanut, M—millet; C2S2, C2S3, C2S4—intercropping of 2 corn rows with 2, 3, 4 rows of soybean, respectively; C2P2, C2P3, C2P4—intercropping of 2 corn rows with 2, 3, 4 rows of peanut, respectively; C2M2, C2M4, C2M6—intercropping of 2 corn rows with 2, 4, 6 rows of millet, respectively.

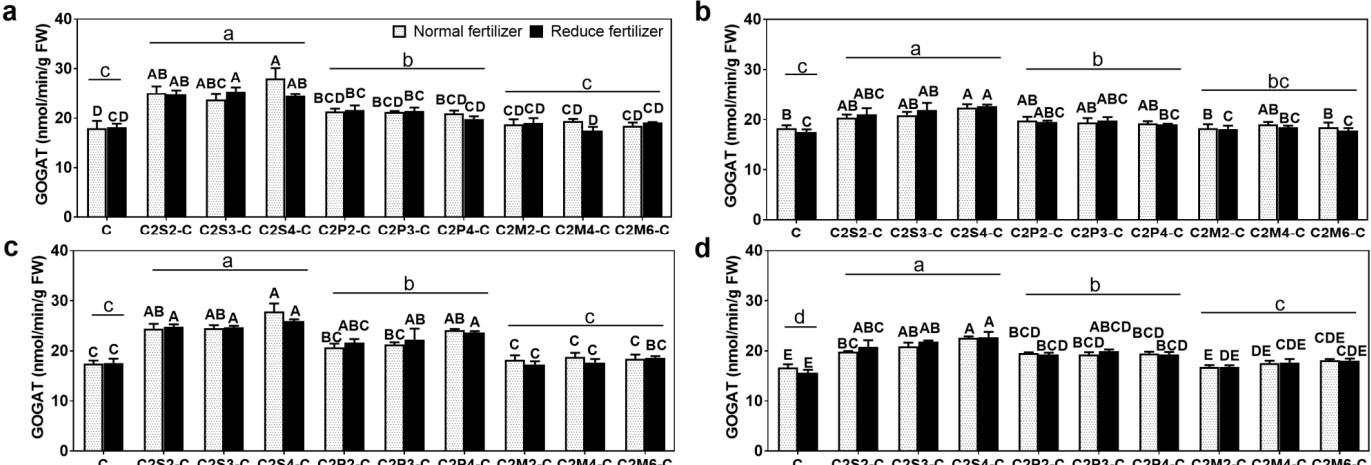

**Figure 6.** GOGAT enzyme activity of corn under different planting patterns in 2018 (**a**,**b**) and 2019 (**c**,**d**): (**a**,**c**)—jointing stage; (**b**,**d**)—dough stage; lowercase letters represent differences between intercropping types; uppercase letters represent differences between planting patterns at the same fertilizer level. C—corn, S—soybean, P—peanut, M—millet; C2S2, C2S3, C2S4—intercropping of 2 corn rows with 2, 3, 4 rows of soybean, respectively; C2P2, C2P3, C2P4—intercropping of 2 corn rows with 2, 3, 4 rows of peanut, respectively; C2M2, C2M4, C2M6—intercropping of 2 corn rows with 2, 4, 6 rows of millet, respectively.

For GS, the enzyme activity under corn-soybean and corn-peanut intercropping was significantly higher than the corn sole crop, and the enzyme activity under corn-soybean was significantly higher than for corn-peanut intercropping. At the corn jointing stage, the activity of GS increased significantly by 61.11–73.01% (corn-soybean, 2018, Figure 7a),

45.49–74.75% (corn-soybean, 2019, Figure 7c), 40.22–47.33% (corn-peanut, 2018, Figure 7a), and 27.88–31.55% (corn-peanut, 2019, Figure 7c) under normal fertilizer compared with the corn sole crop. Under reduced fertilizer application, the activity of S-ALPT increased significantly by 57.49–64.30% (corn-soybean, 2018, Figure 7a), 35.10–51.25% (corn-soybean, 2019, Figure 7c), 35.90–44.05% (corn-peanut, 2018, Figure 7a), and 14.78–27.80% (corn-peanut, 2019, Figure 7c). At the corn dough stage, the activity of S-ALPT increased significantly by 31.44–44.87% (corn-soybean, 2018, Figure 7b), 38.12–52.16% (corn-soybean, 2019, Figure 7d), 13.21–26.37% (corn-peanut, 2018, Figure 7b), and 18.86–33.40% (corn-peanut, 2019, Figure 7d) under normal fertilizer compared with the corn sole crop. Under reduced fertilizer application, the activity of S-ALPT increased significantly by 38.77–54.66% (corn-soybean, 2018, Figure 7b), 47.53–67.62% (corn-soybean, 2019, Figure 7d), 18.88–30.91% (corn-peanut, 2018, Figure 7b), and 28.90–48.69% (corn-peanut, 2019, Figure 7d).

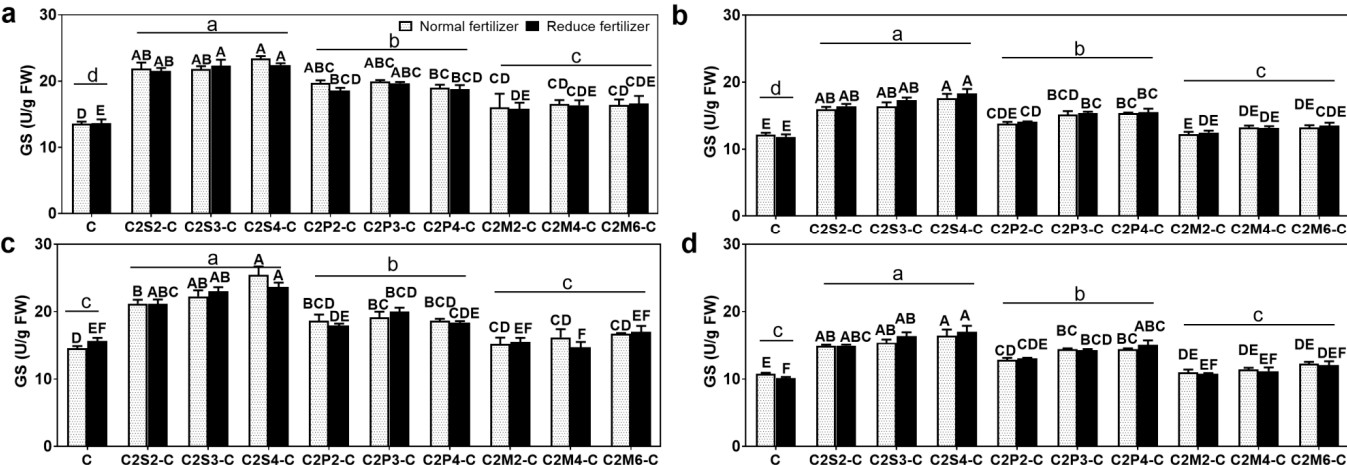

**Figure 7.** GS enzyme activity of corn under different planting patterns in 2018 (**a**,**b**) and 2019 (**c**,**d**): (**a**,**c**)—jointing stage; (**b**,**d**)—dough stage; lowercase letters represent differences between intercropping types; uppercase letters represent differences between planting patterns at the same fertilizer level. C—corn, S—soybean, P—peanut, M—millet; C2S2, C2S3, C2S4—intercropping of 2 corn rows with 2, 3, 4 rows of soybean, respectively; C2P2, C2P3, C2P4—intercropping of 2 corn rows with 2, 3, 4 rows of peanut, respectively; C2M2, C2M4, C2M6—intercropping of 2 corn rows with 2, 4, 6 rows of millet, respectively.

For NR, the enzyme activity of corn-soybean and corn-peanut intercropping was significantly higher than corn-peanut intercropping, and the enzyme activity of corn-soybean intercropping was significantly higher than corn-peanut intercropping. At the corn jointing stage, intercropping with normal fertilizer significantly increased the activity of NR by 120.83–148.61% (corn-soybean, 2018, Figure 8a), 120.00–149.33% (corn-soybean, 2019, Figure 8c), 55.56–59.72% (corn-peanut, 2018, Figure 8a), and 38.67–80.00% (corn-peanut, 2019, Figure 8c), compared with the corn sole crop. Under reduced fertilizer application, the activity of S-ALPT significantly increased by 139.68–168.61% (corn-soybean, 2018, Figure 8a), 131.88–176.81% (corn-soybean, 2019, Figure 8c), 52.38–66.67% (corn-peanut, 2018, Figure 8a), and 57.97–73.91% (corn-peanut, 2019, Figure 8c). Under normal fertilizer, it increased by 92.98–138.56% (corn-soybean, 2018, Figure 8b), 71.79–146.15% (corn-soybean, 2019, Figure 8d), 21.05–89.47% (corn-peanut, 2018, Figure 8b), and 15.38–71.79% (corn-peanut, 2019, Figure 8d) compared with the corn sole crop at the corn dough stage. Under reduced fertilizer application, the activity of S-ALPT significantly increased by 121.57–182.35% (corn-soybean, 2018, Figure 8b), 100.00–177.78% (corn-soybean, 2019, Figure 8d), 31.37–105.88% (corn-peanut, 2018, Figure 8b), and 41.67–72.22% (corn-peanut, 2019, Figure 8d).

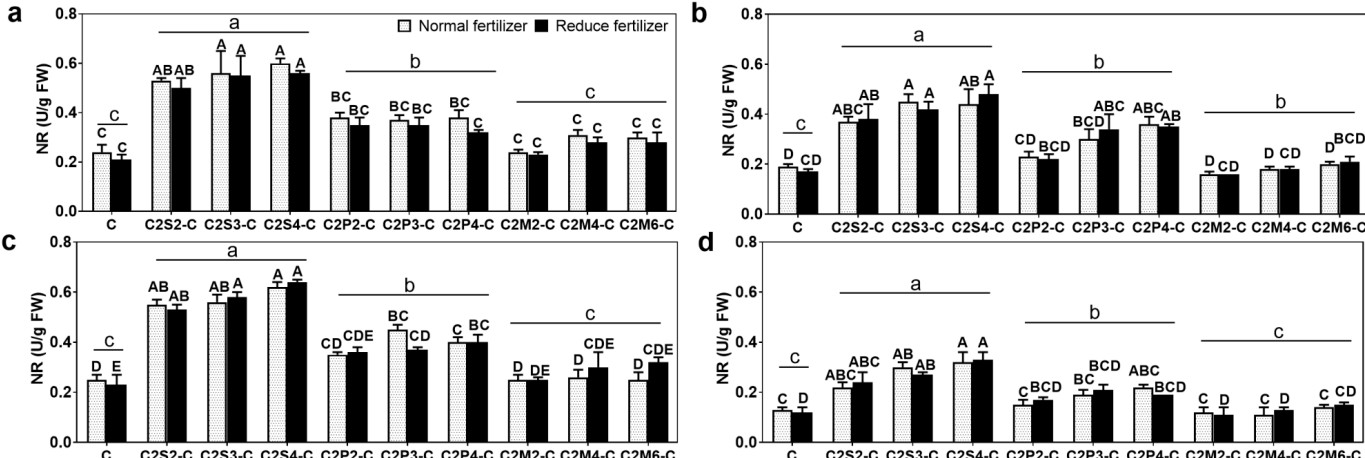

**Figure 8.** NR enzyme activity of corn under different planting patterns in 2018 (**a**,**b**) and 2019 (**c**,**d**): (**a**,**c**)—jointing stage; (**b**,**d**)—dough stage; lowercase letters represent differences between intercropping types; uppercase letters represent differences between planting patterns at the same fertilizer level. C—corn, S—soybean, P—peanut, M—millet; C2S2, C2S3, C2S4—intercropping of 2 corn rows with 2, 3, 4 rows of soybean, respectively; C2P2, C2P3, C2P4—intercropping of 2 corn rows with 2, 3, 4 rows of peanut, respectively; C2M2, C2M4, C2M6—intercropping of 2 corn rows with 2, 4, 6 rows of millet, respectively.

On the whole, the activity of various enzymes increased with the decrease of corn row ratio in planting pattern, and the C2S4-C mode was the best for improving enzyme activity. There were no significant differences in enzyme activities under corn-millet intercropping compared with the corn sole crop (Figures 5–8).

*3.3. Total Nitrogen Content in Rhizosphere Soil and Leaves of Corn*

Analysis of two-year data showed that fertilizer and intercropping type had significant effects on total nitrogen (N) in corn leaves and soil ($p < 0.001$), and the planting pattern had significant effects on total nitrogen ($p < 0.05$, Table 3).

**Table 3.** Three-way ANOVA on the effects of fertilizer level (F; normal vs. reduced, intercropping type (T; corn-soybean/peanut/millet intercropping)), planting patterns (P'; one corn sole crop and three planting patterns of corn intercropping with soybean/peanut/millet), and their bi-/tri-interactions on total N of corn leaf and soil in 2018 and 2019 (*F/p* value).

| Impact Factors | 2018 | | 2019 | |
|---|---|---|---|---|
| | Total N of Leaf | Total N of Soil | Total N of Leaf | Total N of Soil |
| Fertilizer (F) | 198.904/<0.001 *** | 117.823/<0.001 *** | 98.774/<0.001 *** | 97.223/<0.001 *** |
| Intercropping type (T) | 124.388/<0.001 *** | 74.032/<0.001 *** | 118.327/<0.001 *** | 131.245/<0.001 *** |
| Planting pattern (P) | 9.324/<0.001 *** | 9.204/<0.001 *** | 6.974/0.003 ** | 7.589/0.002 ** |
| F × P | 0.054/0.948 | 0.156/0.856 | 0.358/0.701 | 0.199/0.820 |
| F × T | 0.221/0.803 | 2.320/0.111 | 0.718/0.494 | 1.595/0.216 |
| P × T | 1.817/0.144 | 1.997/0.114 | 1.851/0.138 | 2.156/0.092 |
| F × P × T | 1.447/0.236 | 0.216/0.928 | 0.307/0.872 | 1.978/0.116 |

** $p < 0.01$; *** $p < 0.001$.

Compared with the sole crop, the N content in soil of intercropping corn increased by 25.34–27.75% (corn-soybean) and by 13.16–25.31% (corn-peanut) under the normal fertilizer in 2018; it decreased by 29.56–35.53% (corn-soybean) and by 13.62–24.16% (corn-peanut) under reduced fertilizer in 2018 (Figure 9a). Compared with the sole crop, the N content in soil of intercropping corn increased by 24.63–29.66% (corn-soybean) and by 9.39–21.82% (corn-peanut) under normal fertilizer in 2019; it increased by 22.61–32.28% (corn-soybean)

and by 4.37–15.49% (corn-peanut) under reduced fertilizer in 2019 (Figure 9b). There was no significant difference under other intercropping. Under the same planting pattern, compared with normal fertilizer, the reduced fertilizer significantly decreased the soil N content of sole crop and intercropping corn by 6.86–13.57% in 2018 (Figure 9a) and decreased it by 4.95–14.79% in 2019 (Figure 9b).

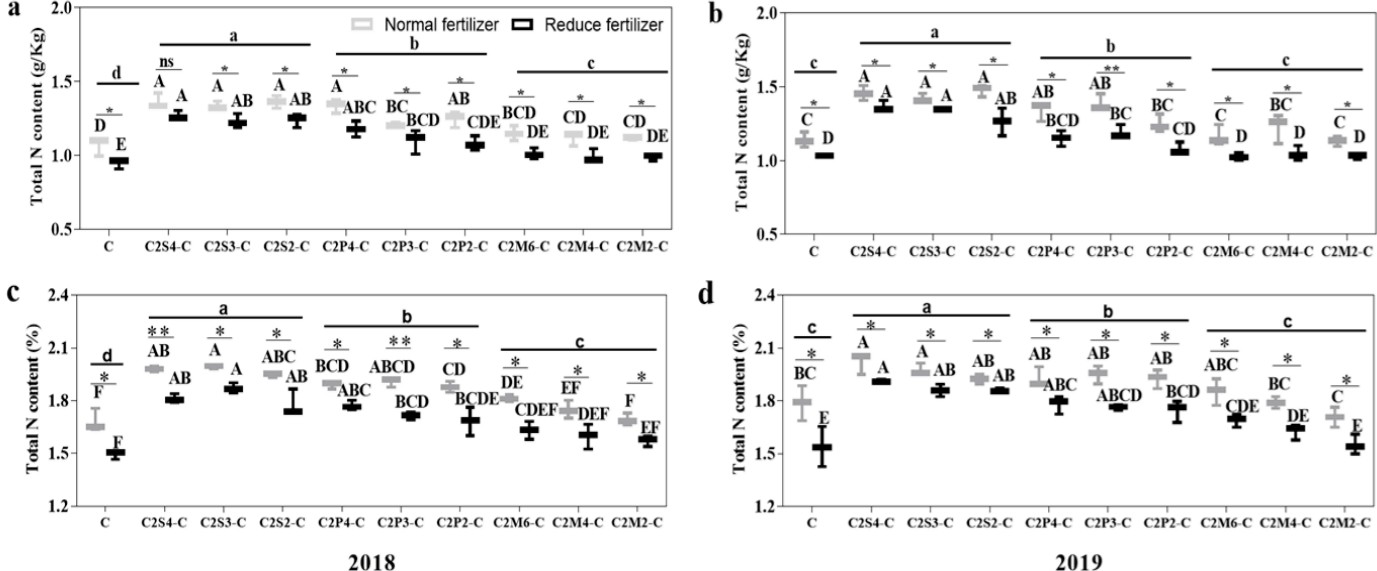

**Figure 9.** Total nitrogen content in rhizosphere soil (**a**,**b**) and leaves (**c**,**d**) of corn under different planting patterns in 2018 (**a**,**c**) and 2019 (**b**,**d**). Lowercase letters represent differences between intercropping types; uppercase letters represent differences between planting patterns at the same fertilizer level; Significant differences between the normal and reduced fertilizer levels in the same planting pattern: * $p < 0.05$; ** $p < 0.01$; ns—not significant, $p \geq 0.05$; C—corn, S—soybean, P—peanut, M—millet; C2S2, C2S3, C2S4—intercropping of 2 corn rows with 2, 3, 4 rows of soybean, respectively; C2P2, C2P3, C2P4—intercropping of 2 corn rows with 2, 3, 4 rows of peanut, respectively; C2M2, C2M4, C2M6—intercropping of 2 corn rows with 2, 4, 6 rows of millet, respectively.

Compared with the corn sole crop, under the normal fertilizer, the total nitrogen content of corn leaves increased by 16.09–18.69% (corn-soybean) and by 11.86–13.50% (corn-peanut) in 2018; it also increased by 18.81–25.14% (corn-soybean) and by 12.62–18.46% (corn-peanut) under the reduced fertilizer in 2018 (Figure 9c). Compared with the corn sole crop, the intercropping increased leaf total nitrogen content by 7.64–13.04% (corn-soybean) and 7.74–9.02% (corn-peanut) under normal fertilizer in 2019, and it increased by 20.71–24.24% (corn-soybean) and 13.38–15.76% (corn-peanut) under reduced fertilizer in 2019 (Figure 9d). Under the same planting pattern, compared with normal fertilizer, the reduced fertilizer decreased the total nitrogen content in corn leaves of sole crop and intercropping by 6.12–10.96% in 2018 (Figure 9c), and decreased it by 3.49–13.94% in 2019 (Figure 9d).

### 3.4. Analysis of Soil Microbial Community

The composition of the soil microbial community in corn rhizosphere under three intercropping types was similar and significantly different from the corn sole crop (Figure 10). The taxonomic composition analysis showed that the number of actinobacteria in rhizosphere soil generally increased under corn intercropping; under corn-soybean intercropping and corn-peanut intercropping, the increase was particularly obvious (Figure 11). Table 4 is an annotation of the taxonomy of soil microbial species under different cropping patterns based on multiple databases.

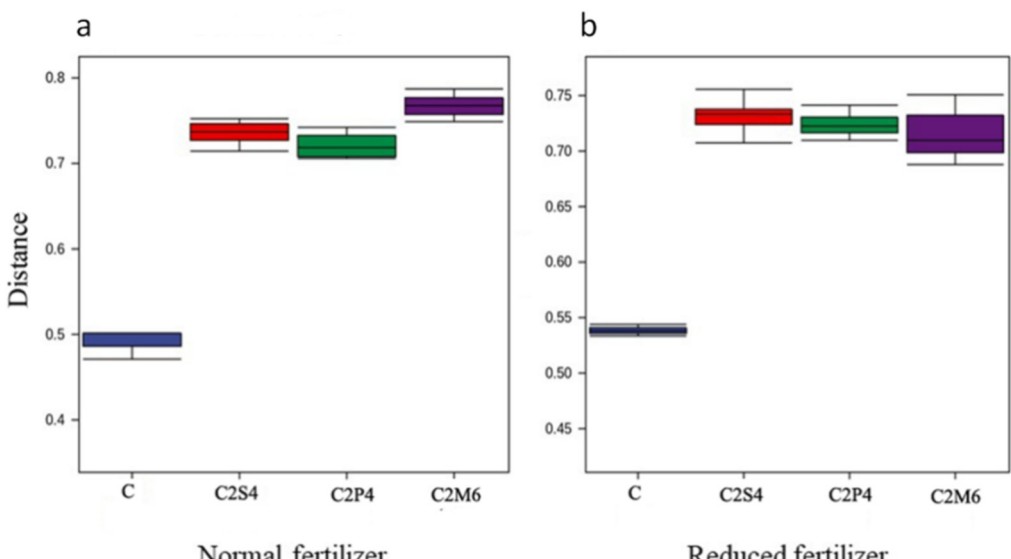

**Figure 10.** Differences of microbial community composition in corn rhizosphere soil under different planting patterns ((**a**)—Normal fertilizer; (**b**)—Reduced fertilizer). C—corn, S—soybean, P—peanut, M—millet; C2S4—intercropping of 2 corn rows with 4 rows of soybean; C2P4—intercropping of 2 corn rows with 4 rows of peanut; C2M6—intercropping of 2 corn rows with 6 rows of millet.

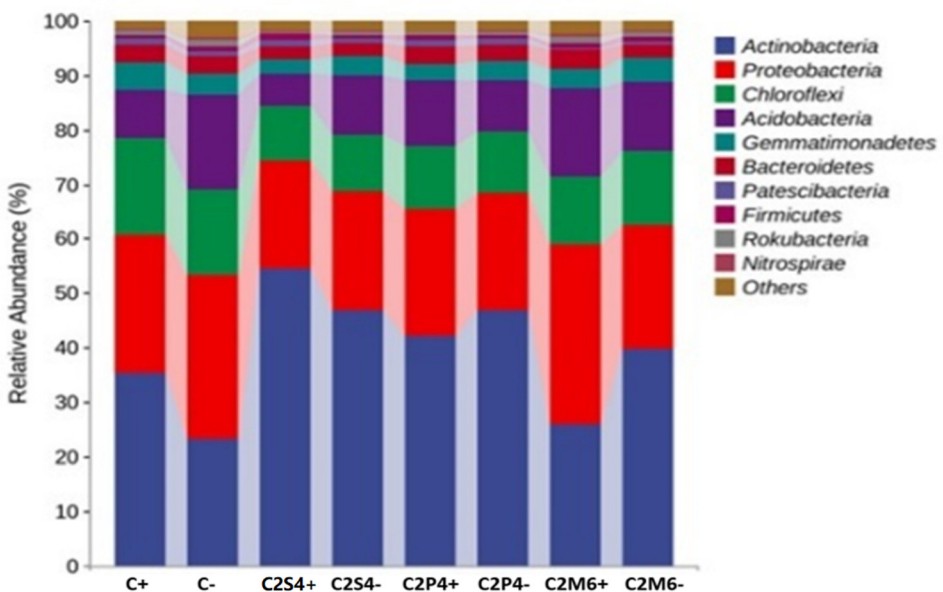

**Figure 11.** Main microbial species under different planting patterns. C—corn, S—soybean, P—peanut, M—millet; C2S4—intercropping of 2 corn rows with 4 rows of soybean; C2P4—intercropping of 2 corn rows with 4 rows of peanut; C2M6—intercropping of 2 corn rows with 6 rows of millet; "+" represents normal fertilizer and "−" represents reduced fertilizer application.

According to two-factor ANOVA analysis of microbial community diversity in corn rhizosphere soil, fertilizer and planting patterns had significant effects on microbial community diversity ($p < 0.05$, Table 5).

**Table 4.** Statistical table of taxonomic annotation results of species.

| Fertilizer Level | Planting Patterns | Domain | Phylum | Class | Order | Family | Genus | Species | Unclassified |
|---|---|---|---|---|---|---|---|---|---|
| Normal fertilizer | C | 66.33 | 32.00 | 169.67 | 241.33 | 864.33 | 3302.33 | 159.00 | 0 |
| | C2S4-C | 76.33 | 48.00 | 215.67 | 316.00 | 978.00 | 3586.67 | 149.67 | 0 |
| | C2P4-C | 66.00 | 39.00 | 194.67 | 299.00 | 953.67 | 3616.33 | 159.33 | 0 |
| | C2M6-C | 101.00 | 50.67 | 197.33 | 280.00 | 926.33 | 3697.67 | 148.67 | 0.33 |
| Reduce fertilizer | C | 114.67 | 46.67 | 164.67 | 244.67 | 955.00 | 3904.00 | 147.00 | 0 |
| | C2S4-C | 78.67 | 62.67 | 237.33 | 307.67 | 1121.33 | 3366.33 | 166.33 | 0.67 |
| | C2P4-C | 122.00 | 53.33 | 218.33 | 277.33 | 1069.00 | 3874.00 | 161.33 | 0 |
| | C2M6-C | 108.33 | 52.00 | 159.33 | 219.67 | 984.00 | 3921.33 | 193.00 | 0 |

C—Corn, S—soybean, P—peanut, M—millet; C2S4—intercropping of 2 corn rows with 4 rows of soybean; C2P4—intercropping of 2 corn rows with 4 rows of peanut; C2M6—intercropping of 2 corn rows with 6 rows of millet.

**Table 5.** Two-factor variance analysis of microbial community diversity index of corn rhizosphere soil in different fertilizer application and planting patterns for Shannon, Pielou-e, and Observed-species (*F/p* value) indices.

| Source of Variation | Shannon | | Pielou-e | | Observed-Species | |
|---|---|---|---|---|---|---|
| | *F* | *p* | *F* | *p* | *F* | *p* |
| Fertilizer (F) | 16.421 | <0.001 *** | 15.076 | 0.0013 ** | 12.363 | 0.003 ** |
| Planting pattern (P′) | 4.674 | 0.016 * | 8.267 | 0.002 ** | 2.482 | 0.098 |
| F × P′ | 2.952 | 0.064 | 2.017 | 0.152 | 2.615 | 0.087 |

* $p < 0.05$; ** $p < 0.01$; *** $p < 0.001$.

Compared with the sole crop, the Shannon index of rhizosphere soil microbial diversity of corn under normal fertilizer significantly increased by 2.32% (C2S4-C), 2.03% (C2S3-C), and 2.23% (C2S2-C) (Figure 12a); the Pielou-e index increased by 1.15% (C2S4-C), 0.90% (C2S3-C), and 0.96% (C2S2-C) (Figure 12b). At the same time, for the sole crop, fertilizer had a significant effect on soil microbial diversity in corn rhizosphere, and fertilizer reduction significantly increased soil microbial diversity. Under fertilizer reduction, compared with normal fertilizer, the soil microbial diversity index of corn rhizosphere soil under the sole crop increased by 2.62% (Shannon, Figure 12a), 0.92% (Pielou-e, Figure 12b), and 15.34% (Observed-species, Figure 12c).

*3.5. Correlation Analysis of Enzyme Activities with Corn Biomass and Yield under Different Planting Patterns*

The correlation analysis of the test results showed that there was a positive correlation between nitrogen metabolism-related enzyme activities and biomass per plant, 1000-grain weight, grain yield, and total N of corn (Figure 13a). Four kinds planting patterns of corn (C, C2S4-C, C2P4-C, and C2G6-C) with the highest yield were selected from each intercropping type, and the correlation analysis of experimental indicators showed that the biomass per plant, 1000-grain weight, grain yield, activities of four nitrogen metabolism-related enzymes, total N, and soil microbial community diversity of corn were also positively correlated (Figure 13b).

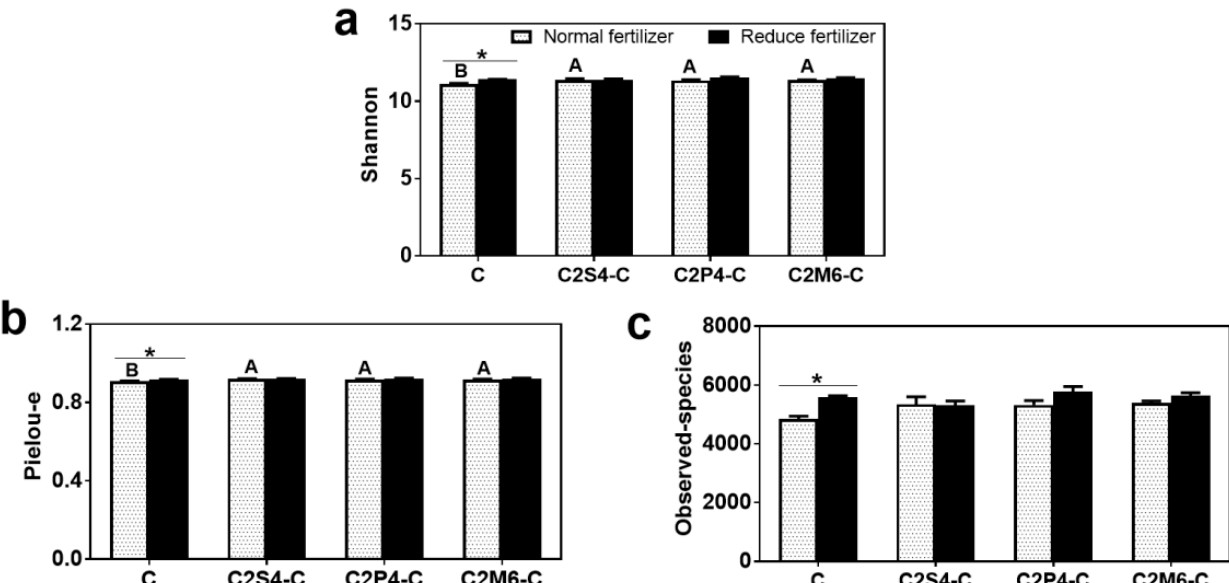

**Figure 12.** Soil microbial community diversity index of corn rhizosphere under different cropping patterns (**a**) Shannon index; (**b**) Pielou-e index; (**c**) Number of observed species; LSD test $p < 0.05$. C—corn, S—soybean, P—peanut, M—millet; C2S4—intercropping of 2 corn rows with 4 rows of soybean; C2P4—intercropping of 2 corn rows with 4 rows of peanut; C2M6—intercropping of 2 corn rows with 6 rows of millet. Uppercase letters represent differences between planting patterns at the same fertilizer level; *—Significant differences between the normal and reduced fertilizer levels in the same planting pattern.

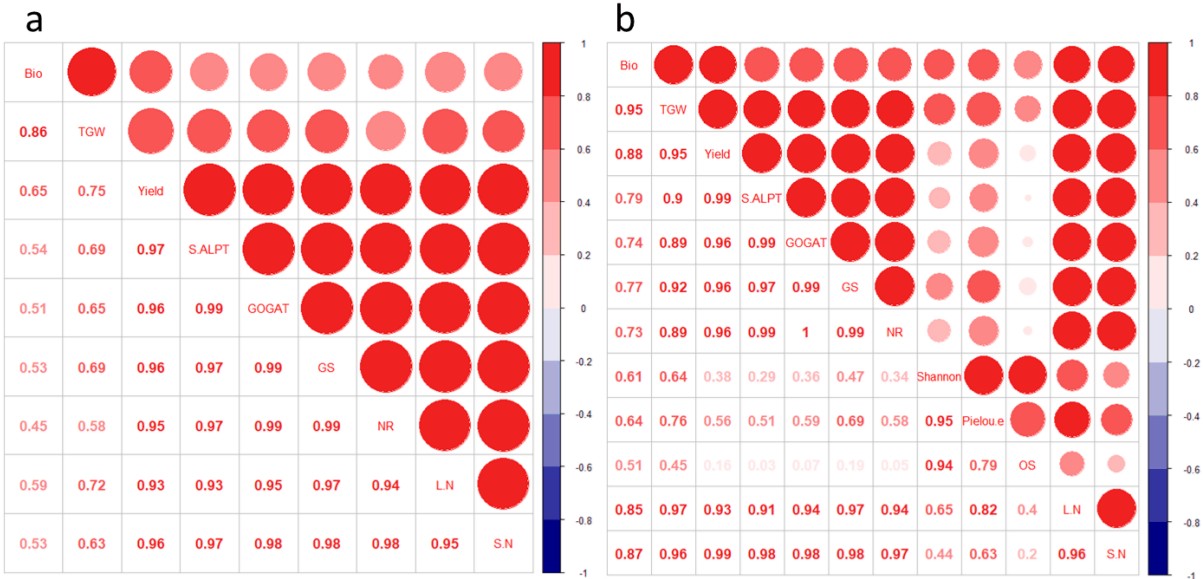

**Figure 13.** Correlation analysis. (**a**) Correlation of biomass per plant (Bio), 1000-grain weight (TGW), and grain yield with enzyme activities related to nitrogen metabolism and total N of leaf (L.N) and soil (S.N) in 2018 and 2019. (**b**) Correlation of biomass per plant (Bio), 1000-grain weight (TGW), grain yield, and enzyme activities related to nitrogen metabolism and total N of leaf (L.N) and soil (S.N) with microbial community diversity of corn (C, C2S4-C, C2P4-C, C2G6-C) rhizosphere soil in 2018 and 2019; S-ALPT—soil alkaline protease; GOGAT—glutamine oxoglutarate aminotransferase; GS—glutamate synthetase; NR—nitrate reductase; OS—Observed-species. The numbers in the picture are correlation coefficients, with red representing positive correlation and blue representing negative correlation. The darker the color, the greater the correlation.

## 4. Discussion

*4.1. Effects of Intercropping Type and Planting Pattern on Plant Biomass and Yield of Corn under Different Fertilizer Levels*

In present study, corn were planted through three types of intercropping with two NFC crops (including soybean and peanut) and one non-NFC crop (millet) through five planting patterns, including three intercropping patterns (2 corn rows to 2, 3, and 4 NFC-crop rows or 2, 4, and 6 millet rows) and sole crop pattern under normal (600 kg/ha) and reduced (375 kg/ha) levels of NPK (N:P$_2$O$_5$:K$_2$O = 15:15:15) fertilizer. Consistent with the results of [55], the biomass per plant and the grain weight per plant of corn in the corn-soybean intercropping and the corn-peanut intercropping were significantly higher than those in the corn-millet intercropping and the sole crop, and the grain weight per plant of corn in the corn-soybean intercropping was slightly higher than that in the corn-peanut intercropping. In the analysis of corn yield, it was found that the fertilizer had no significant effect on corn yield under the intercropping with soybean and peanut. Therefore, the intercropping between corn and NFC crops (e.g., legumes) could ensure grain yield while properly reducing fertilizer, consistent with the results of [56]; NFC crops can increase soil nutrients and promote crop growth. The meta-analysis found that intercropping resulted in higher and more stable crop productivity [29]. According to the experimental data, reducing fertilizer application reduced crop biomass and yield, but the overall difference was not significant. The cost saved by reducing fertilizer application was greater than the yield loss, so it was a more reasonable planting mode from the economic point of view.

The biomass per plant and the grain weight per plant of the intercropping corn were higher than those of the sole crop corn. The competitive effect of corn on other crops under intercropping conditions cannot be ignored: as a tall stalk crop, corn could take some of the nutrients from the intercropped crops because corn has the advantage in space competition aboveground [57]. In addition, in terms of fertilizer utilization, corn also has advantages, and nutrient competitiveness has been significantly improved [58]. The conclusion of this study is inconsistent with that of Dheyme et al. [59]. Dheyme et al. believed that in terms of silage, fermentation method, and nutrient composition, the reduction of nutrient content in corn-soybean intercropping was not conducive to silage production. However, in this study, corn-soybean intercropping could significantly increase corn biomass. In this aspect, it makes up for the economic benefits of nutrition and other aspects, so it is recommended to intercrop corn and soybean.

In addition, we hypothesized that light conditions under the intercropping condition would also have some effects on plant biomass and yield of the intercropping crops; intercropping can improve the utilization rate of light and heat resources [60,61]. Intercropping systems help crops increase capture of light, water, and nutrients and obtain the greatest grain yield [29]. For example, corn-peanut intercropping would increase the photosynthetic rate of corn functional leaves; in contrast, the light compensation point and light saturation point of peanut functional leaves would decrease [62]. Corn is a high-stem crop relative to the intercropping crops of soybean, peanut, and millet, and therefore has a certain shading effect on the intercropped low-stem crops, resulting in the reduction of the photosynthate that forms seeds [63]. Similar conclusions were obtained in corn-wheat intercropping [64]. Intercropping translates intercepted radiation into crop yield more effectively than the sole crop [33]. The light harvesting capacity of intercropping crops is generally higher than that of the sole crop [64]. For example, compared with monoculture, the light harvesting capacity of the corn-wheat intercropping system is 23% higher [65]. However, intercropping patterns have generally had adverse effects, such as shading [66,67], on dwarf crops. Whether the overall benefit of intercropping pattern conforms to actual production needs will be clarified in the next study.

After analysis of two years of data, it was found that crop yields varied in different years. It may be due to the different weather conditions during the growing period of crops in different years. For example, the change of precipitation affects crop yield [68].

*4.2. Effects of Intercropping Type and Planting Pattern on Activities of Enzymes under Different Fertilizer Levels*

It was found that the corn-soybean/peanut (not millet) intercropping with the planting patterns of 2 corn rows to 2, 3, and 4 NFC-crop rows could significantly increase the activities of enzymes related to nitrogen metabolism in the rhizosphere soil (i.e., S-ALPT) and plant leaves (including GOGAT, GS, and NR) of corn under normal and reduced fertilizer, which was consistent with the results of [8,69].

The correlation analysis between yield and enzyme activity showed that there was a significant positive correlation between nitrogen metabolism-related enzyme activity in leaf and soil with corn yield, indicating that corn yield could be increased with the increase in enzyme activity. Corn intercropping with nitrogen fixation crops can significantly increase the related enzyme activity; this study found that the corn-soybean intercropping was favorable to increase enzyme activity and increase the corn yield. Similar results have been obtained from related studies, and reasonable intercropping with legumes can improve the activities of NR and S-ALPT [70]. Wheat and soybean intercropping can increase the acid phosphatase secretion of crop roots [71]. Soybean-sorghum intercropping could increase the activities of dehydrogenase and nitrate reductase in soil [72]. The activities of NR, catalase, and S-ALPT in the rhizosphere soil under corn and peanut intercropping were higher than those under the corn sole crop [69].

Intercropping type and planting pattern had significant effects on the activity of the nitrogen metabolism-related enzymes in the rhizosphere soil (i.e., S-ALPT) and plant leaves (including GOGAT, GS, and NR) of corn. The enzyme activities of S-ALPT, GOGAT, and GS in the corn of corn-soybean intercropping and corn-peanut intercropping were significantly higher than those in the corn-millet intercropping, especially under reduced fertilizer, and the activities of the three enzymes in the corn-soybean intercropping were higher than those in the corn-peanut intercropping (Figure 4). Therefore, soybean is the optimal NFC crop; it has stronger nitrogen fixation capacity than the NFC crop of peanut [73], which is preferred and generally used as a mode of green technology in intercropping system to increase crop production; other studies have also shown this [74].

It was concluded that the corn-soybean intercropping with 2 corn rows to 4 soybean rows was the best intercropping type and planting pattern to improve the activities of the nitrogen metabolism-related enzymes in the rhizosphere soil (i.e., S-ALPT) and plant leaves (including GOGAT, GS, and NR) of corn, especially under reduced fertilizer. Some enzyme activities in tea plants [75] and corn [76] decreased with the increase in leaf age, which was consistent with the results of the present experiment. In this study, it was found that a reasonable intercropping pattern could guarantee yield under the condition of reduced fertilizer application, which was supported by a large number of relevant studies in previous reviews [77]. The yield of silage could also be increased by reducing nitrogen fertilizer input in corn-alfalfa intercropping [78].

*4.3. Effects of Intercropping Type and Planting Pattern on Soil Microbial Community under Different Fertilizer Levels*

Some studies have shown that intercropping is beneficial to the growth and reproduction of microorganisms in the field and can increase the diversity of soil microorganisms; the activity of microorganisms in the rhizosphere can stimulate plant growth, promote the production of biomass, and promote the release of organic compounds derived from photosynthesis to the rhizosphere, thus supporting the growth and activity of microorganisms [39], consistent with the present study.

Correlation analysis showed that there was no significant correlation between soil microbial diversity and corn grain yield. However, the changing trend of corn grain yield and soil microbial diversity was consistent under intercropping mode, indicating that the increase of microbial diversity had a positive effect on improving corn grain yield. In addition, reduced fertilizer application can increase soil microbial diversity (Figure 9), which may be due to the inhibitory effect of chemical fertilizers on soil microorganisms.

Studies have found that too much nitrogen input can lead to potentially toxic concentrations [79,80]. It has an inhibitory effect on soil microorganisms [81–83]. Compared with the corn sole crop, corn rhizosphere soil microorganisms under normal fertilizer were significantly increased under intercropping, which was expected because corn used the excess fertilizer, thus relieving the pressure of soil microorganisms, but the actual situation needs to be verified by experiments.

## 5. Conclusions

In the present experiment, the results showed that the activities of N-metabolism-related enzymes (including S-ALPT, GOGAT, and GS) and the grain weight per plant and grain yield of corn intercropped with soybean and peanut were significantly increased with the increase of the row ratio of soybean and peanut intercropping with corn, especially in the planting pattern of 2 corn rows to 4 soybean/peanut rows (i.e., C2S4 and C2P4), even under reduced fertilizer application. The grain weight per plant and grain yield of corn in the corn-soybean intercropping was higher than that of corn in the corn-peanut intercropping. Corn intercropping with soybean and peanut increased the grain yield and enzyme activities related to nitrogen metabolism of corn. The mechanism of intercropping promoting the increase of grain yield might be that the enzyme activities related to nitrogen metabolism of corn were increased for nitrogen-fixing crops, and then the biomass per plant was increased, and finally, the harvest grain yield was increased. It is concluded that corn intercropping with the NFC crops must be better than that intercropping with non-NFC crops, and the corn-soybean intercropping is the best intercropping type.

**Author Contributions:** L.L.: Conceived and designed the experiments, performed the experiments, analyzed the data, prepared figures and/or tables, authored or reviewed drafts of the paper, approved the final draft. Y.Z.: Performed experiments and helped with data analysis. Y.W.: Performed experiments and helped with data analysis. F.C.: Conceived the study, participated in its design and coordination, helped to revise the manuscript, and approved the final draft. G.X.: Conceived the study, participated in its design and coordination, helped to revise the manuscript, and approved the final draft. All authors have read and agreed to the published version of the manuscript.

**Funding:** This research was funded by the National Key Research and Development Program of China (2017YFD0200400) and Jiangsu JCIC-MCP Program.

**Institutional Review Board Statement:** Not applicable.

**Informed Consent Statement:** Not applicable.

**Data Availability Statement:** Not applicable.

**Conflicts of Interest:** The authors declare no conflict of interest.

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
