# Peer review of "Effects of Corn Intercropping with Soybean/Peanut/Millet on the Biomass and Yield of Corn under Fertilizer Reduction"

_agriculture, doi:10.3390/agriculture12020151_

Round 1

Reviewer 1 Report

The manuscript is good. But as for the characteristics of biomass, the literature review can be developed. The authors should take into account similar studies that have been carried out in the discussion. The authors described the subject of biomass. I recommend that you refer to biomass in more chemical aspects, I recommend reading the article: https://doi.org/10.3390/pr9020364 .For example in the above article, the PY-GC-MS of straw was shown.

It would be also good to describe the economic impact of the technology used and compare how much the commonly used technology (an example estimate by the used fertilizers) would cost to that proposed by the authors' technology. This is important because the cost of implementing the technology is the basis for its application. 

Reviewer 2 Report

The topic of this experimental paper is well-chosen, topical, and within Agriculture. However, it needs refinement and thorough revision. 
The information presented is not truly complete and needs to be updated. Greater attention should be paid to the following points:
- The current version lacks a critical review and elaboration of the known literature, merely repeating what has already been presented. An excellent experimental article (especially its discussion) should not only report what has already been written in the past. Still, it should critically evaluate the literature and compare it with the results obtained, especially where there are conflicting reports.

The organization of the manuscript is somewhat unorganized, and some of the figures are not very useful in their present form. The results section should be more concise and more elaborate overall. 
The figures do not really summarise the state of the findings, are not illustrative, and some are not very conclusive.
References should be double-checked, and formatting should be done strictly according to journal rules. 
Further, typos were consistently found in the ms, and authors are asked to read their ms carefully before submitting. 
Below, I will address some of the other points I found while reviewing the ms:

  • I would consider changing the title- it's a bit misleading, and it's more of a classic nutritional experiment than a focus on intercropping
    - the introduction lacks more up-to-date literature (only one publication from 2021) focusing on intercropping in the plants studied (soybean, corn, millet)
  • Add some recent references in the MS. It is important to discuss plant regulation and yield mechanisms. The paper brings many new aspects, and the paper's novelty is OK. Still, I would like to invite authors to discuss also more eco-physiological aspects using new references: 10.1111/grs.12323; 10.1071/FP19161; 10.1111/gcb.15747; 10.3390/agronomy11020343; 10.1016/j.plaphy.2020.11.053; 10.1016/j.fcr.2021.108068
  • Is the initial nutrient supply of the experimental soil known?
  • Have you tried using another statistical test? HSD Tukey - would be more rigorous and show clear significance.
  • Fig. 2 and 7 are incorrectly embedded - not showing the whole thing, labeling of experimental variants is not clear - as is the labeling of conclusiveness - would recommend improving and using color to increase the citability of the paper.
  • General captions for figures need to be reworked - each figure or table needs to be self-supporting.
  • Chapter. 2.5 - is this a new method of soil microbial analysis? No citation is given. At the same time, these results are the least conclusive - see fig. 12.

Reviewer 3 Report

The paper, titled “Effects of different intercropping systems on the biomass and yield of corn under fertilizer reduction” by Likun Li et al., is very interesting and the content is based on reliable data. Although the authors did an excellent job on both the field and experimental parts of the paper, some improvements would make the paper more relevant to the scientific community.

Page 1, line 11 Next to the name of the crop in English, write the scientific name in Latin, for example, Corn (Zea mays L.), soybean (Glycine hispida L.), etc. Do the same for other crops below.

Page 1, line 19 instead of N: P2O5: K2O=15: 15: 15 write N: P2O5: K2O=15: 15: 15. Apply the same subscripts in the text below.

Page 1, line 30 Sentences that begin with And rewrite or begin with furthermore, or in addition depending on the context of the sentence. Correct such sentences throughout the text.

Page 3, line 112-119 The sentence is too confusing. Please make to clearly state what are the main goals of the existing research. Explain in detail the setting of the experiment in the material and methods, and focus and explain more the hypotheses of your research.

Page 3, line 128 Soil type is fluvo-aquic soil – based on which soil classification?

Page 3, line 147 Please specify whether the amount of applied mineral fertilizers (normal and reduced) was determined based on soil analysis conducted before the experiment or was it determined using some other assumption? It is unclear whether or not the amount of mineral fertilizer applied is the same for all crops since not all crops have the same fertilization needs. Please clarify when the mineral fertilizer was used (as the total amount applied as basic fertilization, before sowing or later?). Please explain in more detail.

Page 4, line 166 Please rewrite the title of section 2.4. „Crop planting and sampling collection” because it also describes laboratory methods for determining enzyme activity or describe that part and change its title in section 2.5. “Measurement and analysis of soil microbial composition”.

Page 4, line 184-185 Moreover the methods used to determine enzyme activity (GOGAT, GS, NR, S-ALPT) should be explained.

Page 5, line 195-223 in this section 2.5. Measurement and analysis of soil microbial composition, add and describe the DNA extraction procedure and how the 16s, 18s, and ITS sequences were analyzed (which PCR protocol was used for the amplification). 

In order to better explain, I recommend checking the following paper:
https://doi.org/10.3390/agronomy11081539

Page 5, line 217-219 Based on your sentence: According to the species composition and distribution in each sample, the association network was constructed to calculate the topological index, in order to find the key species. In your results, you didn't write which key species you got. Please provide.

Page 5, for the whole section 3. Results It is redundant to put in parentheses for each result where a table is shown (unnecessary text), it might be easier to mention at the beginning of each section that the given results are shown in Table specific number. The exception should be made when entering data for which there is a new Table or Figure.

Page 6, Table 1 Yield (kg/hm2)? You should write kg/ha.  Do the same in Figure 6.

Please format the figures properly. There are some figures that are not properly visible. For example Figure 7,8,9,10.

The explanation should specify in which unit of measurement enzyme activity is measured, as shown for the other parameters. It should be presented more clearly on the figure, or defined in the figure caption.

Page 18-19 3.3 Soil microbial diversity of corn under different cropping patterns. It is necessary to add in this section display of a representation of filum (based 16s, 18s, and ITS) before Figure 11.

Page 20, Figure 12.  line 27, Did you mean Duncan’s test?

Page 21, line 92-93 After two-year of data analysis, it was found that crop yields varied in different years, possibly due to climate changes between years. Perhaps it is more correct to say due to the different weather conditions between the years.

Arrange references according to the instructions for authors (MDPI template).

Round 2

Reviewer 2 Report

The authors have significantly improved the manuscript and addressed all the reviewers' comments. The manuscript can now be recommended for publication.

Author Response

Thank you very much for your contribution to this manuscript.

Reviewer 3 Report

Thank you for completing the revision and for an explanation of the raised issues. There is just one more thing which should be better explained:

In Section 2.4. Sample collection and determination, before line 187 should be added:

PCR protocol was used for the amplification of specific DNA fragments (for 16S, 18S and ITS region), and Kit was used for the isolation of total genomic DNA.

In order to better explain what you should do, I recommend checking the following papers:

https://doi.org/10.3390/agronomy11122425  (See section 2.3. in the recommended paper for used kit for DNA isolation) and https://doi.org/10.3390/agronomy11122400 (See section 2.3.1. in the recommended paper for PCR protocol used for the amplification of specific DNA fragments (for 16S, 18S and ITS region).

Author Response

Thank you very much for your contribution to this manuscript. At the same time, thank you for the literature you recommended.

The related content (PCR protocol was used for the amplification of specific DNA fragments (for 16S, 18S and ITS region), and Kit was used for the isolation of total genomic DNA.) has been added before line 187, please review the manuscript again..